# Adversarial Experts Model for Black-box Domain Adaptation

## ABSTRACT

Black-box domain adaptation treats the source domain model as a black box. During the transfer process, the only available information about the target domain is the noisy labels output by the black-box model. This poses significant challenges for domain adaptation. Conventional approaches typically tackle the black-box noisy label problem from two aspects: self-knowledge distillation and pseudo-label denoising, both achieving limited performance due to limited knowledge information. To mitigate this issue, we explore the potential of off-the-shelf vision-language (ViL) multimodal models with rich semantic information for black-box domain adaptation by introducing an *Adversarial Experts Model (AEM)*. Specifically, our target domain model is designed as one feature extractor and two classifiers, trained over two stages: In the *knowledge transferring* stage, with a shared feature extractor, the black-box source model and the ViL model act as two distinct experts for joint knowledge contribution, guiding the learning of one classifier each. While contributing their respective knowledge, the experts are also updated due to their own limitation and bias. In the *adversarial alignment* stage, to further distill expert knowledge to the target domain model, adversarial learning is conducted between the feature extractor and the two classifiers. A new consistency-max loss function is proposed to measure two classifier consistency and further improve classifier prediction certainty. Extensive experiments on multiple datasets demonstrate the effectiveness of our approach. Our source code will be released.

## CCS CONCEPTS

• **Computing methodologies** → **Transfer learning**; **Neural networks**.

## KEYWORDS

Black-box domain adaptation, adversarial Learning, vision-language pre-trained model

**ACM Reference Format:**
Anonymous Author(s). 2024. Adversarial Experts Model for Black-box Domain Adaptation. In *Proceedings of xxx (Melbourne'24)*. ACM, New York, NY, USA, 9 pages. https://doi.org/XXXXXXX.XXXXXXX

## 1 INTRODUCTION

Unsupervised domain adaptation involves transferring knowledge from a labeled source domain to an unlabeled target domain. The goal is to achieve excellent performance for the target model [9,

(a) Previous black-box domain adaptation methods

(b) The proposed AEM (our method)

Figure 1: (a) Previous methods use noisy labels generated from black-box model for self-supervised learning in the target domain. (b) Our approach leverages and updates the knowledge from two experts (the black box and ViL models). Adversarial learning is employed to distill knowledge to the target domain from two experts.

25]. For the sake of data security and personal privacy protection, source-free domain adaptation is proposed, which only employs the model from the source domain and data from the target domain for training [20, 39]. In fact, the data distribution of source domain can be estimated by the information carried on the source model, which also implies the risk of data leakage [21, 28]. Moreover, providing AI services through cloud API has become a future development trend. Therefore, black-box domain adaptation [21] has emerged, treating the source domain as a black box. Then, using the labels output by the black-box model, source information is transferred to the target domain model.

The current research on black-box domain adaptation is limited. The closely related unsupervised domain adaptation methods consist of three technical routes: traditional distribution alignment [14, 25, 27], adversarial learning [2, 9, 23], and self-supervised learning [8, 12, 37]. Inspired by the works on unsupervised domain adaptation problem, there are also three main approaches on source-free domain adaptation [17], i.e, generative learning [5, 18, 40], distribution alignment [7, 24] and self-supervised learning [4, 15, 19, 20]. The above mentioned approaches cannot be applied to the problem of black-box domain adaption directly, because both of source data and model are not accessible.

Typically, due to the distribution difference between the source and target domains, the labels output by the black-box model are

noisy. This can lead to negative transfer to the target domain. Consequently, there are two types of black-box domain adaptation methods to solve this problem. The first category is self-distillation learning through knowledge distillation of target data [21, 28]. The second category is pseudo-label denoising learning, utilizing the target data to get denoised labels [43, 44]. All of the above methods only exploit the information inherent in the target domain data, resulting in the target domain model lacking high-level semantic information. The performance hits ceiling. Therefore, introducing additional supervision information is an intuitive idea. Vision-language (ViL) multimodal models (e.g., CLIP [30]) align text and images in a high-level semantic space using contrastive loss, which is a good auxiliary model to help adaptation. Unfortunately, directly using ViL model is not a good idea because it has error labels for some specific domains. Moreover, simply aligning the target domain model to the black-box and the ViL model labels is also not wise. Because the knowledge learned by these two models is heterogeneous, there are inconsistent labeled target samples.

Based on the aforementioned observation, we propose a novel **adversarial experts model (AEM)** for black-box domain adaptation. In our method, the target domain model is designed as a shared feature extractor and two classifiers. The two classifiers are dominated respectively by the black-box expert and the ViL model expert. The overall framework of AEM is divided into two stages, namely *knowledge transferring* and *adversarial alignment*. In the *knowledge transferring* stage, the pseudo-labels of overall target domain samples from both the black box and the ViL model are used for supervised learning. As the expert also has limitations, the expert knowledge is updated based on the characteristics of the target domain in each epoch. In the *adversarial alignment* stage, the difference between the predictions from two classifiers is maximized in the hope of identifying hard samples. And then, the classification difference is minimized to update the feature extractor, expecting to extract consistency features for two classifiers. Through adversarial learning between the feature extractor and the two classifiers, the target domain model gradually consolidates and aligns the expert knowledge. The learned features are more discriminate and domain invariant. Additionally, we propose a new classifier Consistency-Max (CM) loss that, while ensuring prediction consistency and certainty, forces the output predictions to be more confident. Extensive experiments demonstrate the effectiveness of AEM, which even rivals the state-of-the-art unsupervised domain adaptation methods using ViL model.

Our contributions can be summarized as follows: (1) We introduce a novel framework for adversarial learning in black-box domain adaptation, leveraging the heterogeneous knowledge from two experts. This approach enriches the target domain model with multifaceted knowledge, ensuring accurate information transfer. (2) Updates are made to the black-box labels and the prompt of the ViL model. Unlike prior methods that solely rely on self-distilled black-box labels, we refine the two experts' predictions. The knowledge from two experts will gradually evolve to better suit the target domain. (3) We propose a novel CM loss to address certainty limitations in previous approaches. Compared with previous methods using metrics like $L_1$ norm or Kullback-Leibler divergence to measure prediction distance, simple yet effective CM loss not only measures prediction consistency but also improves prediction certainty.

## 2 RELATED WORKS

**Domain adaptation.** *Unsupervised domain adaptation* (UDA) can be roughly divided into three categories: distribution alignment, self-supervised learning and adversarial learning. Despite achieving good results, UDA methods face privacy leakage issues. Additionally, due to end-to-end supervised training from the source domain, the resulting target domain model exhibits source bias. This implies that the final model inherently carries negative transfer problems. *Source-free domain adaptation* (SFDA) relies solely on pre-trained source models without access to the source data. There are three main types of existing source-free domain adaptation methods [17]. The first category involves generative learning [5, 18, 40], where the goal is to create source-related data to align the target domain with the source domain. The second approach relies on aligning distributions [7, 24]. Batch statistics stored within a pre-trained source model is leveraged to approximate the distribution of inaccessible source data. Subsequently, cross-domain adaptation is achieved by directly minimizing the distribution gap between the source and target domains. The third category of methods is based on self-supervised learning, including self-supervised knowledge distillation [19, 20], pseudo-labeling denoising [4, 15] and so on. Source-free domain adaptation approaches also can not be applied directly to the black-box domain adaptation because the source model is unknown.

**Black-box domain adaptation.** Existing black-box domain adaptation methods can be summarized into two approaches. The first approach is *self-distillation learning*, aiming to gradually adapt models or labels using a teacher-student architecture [21, 28]. Symbolically, DINE [21] takes the pesudo-labels from the black-box model as the teacher, gradually distilling knowledge to the target domain model. The second approach is based on *pseudo-label denoising*, which leverages information from the target domain data to alleviate label noise in black-box model outputs [43, 44]. For example, BiMem [43] employs a bi-directional memory module to store useful information of long-term and short-term contexts, thereby alleviating the issue of noisy labels in black-box scenarios. Although these methods have achieved good results, they only utilize noisy label information from the black box and information from the target domain itself, failing to capture high-level semantic information. This imposes performance limitation on the target domain model.

**Vision-language multimodal model.** Vision-language (ViL) multimodal models have achieved remarkable performance in various fields. Through training on massive datasets, ViL models effectively capture the connections between different modalities of data and derive output results from a higher semantic level. Current research on ViL models can be broadly categorized into two main streams. The first stream focuses on model optimization, exploring ways to train ViL models with higher accuracy or reduced resources [16, 45]. For example, BLIP [16] harnesses noisy web data through caption bootstrapping, which advances the performance of many vision-language tasks. The second stream delves into downstream adaptation researches [3, 41]. For example, USL-VI-ReID [3] establishes a trainable cluster-aware prompt, which acquires textual descriptions that facilitate subsequent unsupervised training. We also apply ViL model to black-box domain adaptation. Instead of

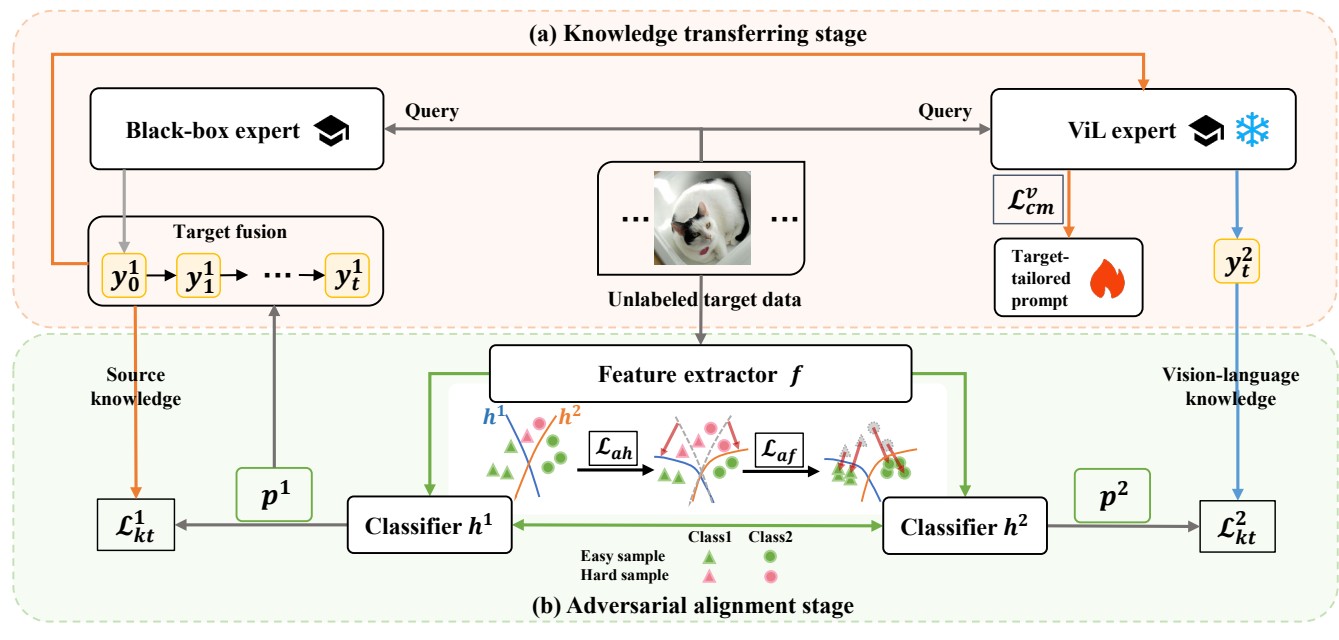

Figure 2: Overall framework of the proposed AEM. The method is divided into two stages: knowledge transferring and adversarial alignment. In the knowledge transferring stage, we utilize two experts for supervised learning in the target domain. Additionally, the experts' knowledge is adaptively adjusted based on target domain information. In the adversarial alignment stage, adversarial learning is employed between the feature extractor and two classifiers to come to a consensus on the hard samples and align the knowledge from two experts.

naively using ViL model, the black-box model and the ViL model are considered as two experts. Adversarial learning is applied to combine and fine-tune expert knowledge.

## 3 PROPOSED METHOD

In the setting of black-box domain adaptation, the labeled source domain data and the source domain model are inaccessible, which is treated as a black-box model. The only information available from the source domain is the noisy labels of the target samples obtained from the black-box model, where the target domain data, consisting of $N$ samples, is denoted as $X = \{(x_i)\}_{i=1}^N$, and the noisy labels obtained from the black-box model are denoted as $Y^1 = \{(y_i^1)\}_{i=1}^N$. The target domain shares the same class space $C = \{1, 2, ..., K\}$ as the source domain. The goal is to train a high-performance target domain model using the information mentioned above.

**Overview.** As depicted in Fig.2, the proposed AEM framework comprises two experts: the black-box expert bringing source domain knowledge and the ViL model expert containing high-level semantic knowledge. Each expert guides the learning of a classifier, denoted as $h^1, h^2$ respectively while sharing a common feature extractor $f$. The overall process is divided into two stages, namely knowledge transferring stage and adversarial alignment stage. In the knowledge transferring stage, the two classifiers are trained under the guidance of their respective experts, while the feature extractor is trained under the joint guidance of both experts. Furthermore, AEM adapts the knowledge of two experts to suit the target domain at each epoch. In the adversarial alignment stage, we

customize a Consistency-Max (CM) loss to ensure the consistency and certainty of the predictions from two classifiers. Utilizing the proposed loss, we fix the feature extractor and maximize the prediction discrepancy to update the two classifiers. Then, the classifiers are fixed and the prediction discrepancy is minimized to update the feature extractor. In this way, two expert knowledge are aligned to the target domain.

### 3.1 Knowledge transferring

The black-box model possesses rich source domain knowledge. However, as the categories in the source and target domains remain the same while their distributions differ, the pseudo-labels obtained from the black-box model for target domain often carry more noise, resulting in noisy pseudo-labels. These noises introduce uncertainty and erroneous signals into the training data, potentially leading the model to learn incorrect representation and causing negative transfer. To address these challenges, we propose introducing a ViL model expert, such as CLIP [30] which incorporates high-level semantic information. ViL models learn rich semantic representations from both visual and textual contexts through contrastive learning, thereby enhancing the robustness of feature representations. However, it's worth noting that ViL models may exhibit sub-optimal performance in specific domains or tasks. Therefore, we suggest that the target domain model learns from both black-box and ViL experts. Successful knowledge transferring from these experts is guaranteed by the knowledge dissemination strategy and knowledge feedback strategy, which will be detailed later.

**Knowledge dissemination strategy.** To start with, the outputs from both experts are used for preliminary pseudo-label supervised learning. Let $Y^1 = \{(y_i^1)\}_{i=1}^N$ denotes the set of soft pseudo-labels output by the black-box model, and $Y^2 = \{(y_i^2)\}_{i=1}^N$ denotes the set of soft pseudo-labels output by the ViL model. And then the corresponding one-hot label sets can be obtained, denoted as $\hat{Y}^1 = \{(\hat{y}_i^1)\}_{i=1}^N$ from black-box and $\hat{Y}^2 = \{(\hat{y}_i^2)\}_{i=1}^N$ from ViL experts, respectively. Next, we compute cross-entropy loss separately between the predictions of the two classifiers $h^1$, $h^2$ and the one-hot labels obtained from black-box and ViL experts respectively, written as follows:

$$\mathcal{L}_{ce}^j = -\mathbb{E}_{x_i \in X}(\hat{y}_i^j \log(p_i^j)), \tag{1}$$

where $p_i^j = h^j(f(x_i))$, $j \in \{1, 2\}$, is the softmax predictions by target model.

As the target domain shares the same class space as the source domain, it implies that the information contained in the target domain is more relative to the black-box expert. Therefore, we further constrain the output of classifier $h^1$ to be consistent with the black-box expert by maximizing mutual information loss [20], which is defined as follows:

$$\begin{aligned}
\mathcal{L}_{mi} &= M(Y) - M(Y|X) \\
&= m(\mathbb{E}_{x_i \in X} h^1(f(x_i))) - \mathbb{E}_{x_i \in X} m(h^1(f(x_i))),
\end{aligned} \tag{2}$$

where $m(p) = -\sum_{k=1}^K p_k \log p_k$ is the entropy function. This term forces the black-box guided classifier to produce unambiguous predictions and encourages the label distribution to be uniform.

Based on the above analysis, we update the feature extractor $f$ and the classifier $h^1$ guided by the black-box expert. To enhance generalization, the widely used MixUp [28, 42] strategy is integrated into the objective function:

$$\min_{f, h^1} \mathcal{L}_{kt}^1 = \mathcal{L}_{ce}^1 - \mathcal{L}_{mi} + \mathcal{L}_{mixup}. \tag{3}$$

Since the ViL model is trained on massive data, it is well at generalization. Therefore, for the classifier $h^2$ guided by the ViL expert, we simply use cross-entropy for straightforward training:

$$\min_{f, h^2} \mathcal{L}_{kt}^2 = \mathcal{L}_{ce}^2. \tag{4}$$

Through the training based on Eqs.(3) and (4), we obtain a shared feature extractor and two classifiers guided by two different experts, respectively. By this process, the target domain model acquires the ability to classify easy samples under the guidance of two experts.

**Knowledge feedback strategy.** Although the experts impart knowledge to the target domain model, their expertise lies outside the target domain. For instance, the knowledge from the black-box model is biased towards the source domain, while the knowledge from the ViL expert is highly generalized and not fine-tuned. In order to achieve better performance in the target domain, the knowledge from both experts should be promptly updated based on the target domain. Therefore, we adapt the knowledge of both experts to be suitable for target domain at each epoch.

Since the black-box model only outputs noisy labels for the target domain and cannot be adjusted, we use the EMA (Exponential Moving Average) strategy to update its pseudo-labels:

$$y_i^1 \leftarrow \alpha y_i^1 + (1 - \alpha) h^1(f(x_i)), \tag{5}$$

where $\alpha$ is a hyperparameter to adjust the update rate. Differing from the sealed and non-fine-tunable black-box model, we hope to continuously learn the high-level semantic information of ViL expert. Thus we are committed to adapting the ViL expert to the target domain. Inspired by CoOp [45], with the ViL model frozen, we utilize black-box labels to fine-tune the prompt of the ViL expert, enabling its knowledge to be made use of in the target domain. Specifically, a consistency-max (CM) loss, introduced and analyzed in Section 3.2, is used to update the prompt:

$$\min_w \mathcal{L}_{cm}^v = -Y^1 \cdot Y^2, \tag{6}$$

where $w$ is the prompt of ViL model to be fine-tuned.

Using the EMA strategy, the black-box labels are gradually replaced by the outputs of $h^1$, thereby incorporating more information from the target domain. The noise issue of the black-box output labels will also be alleviated. From the perspective of knowledge distillation, the black-box labels serve as the teacher's role in the teacher-student model, carrying more authoritative information. Therefore, we use the updated black-box labels to fine-tune the prompt of ViL. Through this process, the output of ViL will also be more suitable for the target domain model.

## 3.2 Adversarial alignment

Through knowledge transferring, the target domain model has learned heterogeneous knowledge from two experts. It means the knowledge learned by the two classifiers is unrelated to some extent. That is, for the same sample, the two classifiers may produce inconsistent results. Additionally, since only one feature extractor is used for learning knowledge from two heterogeneous spaces, the extracted features are likely to lack discrimination. So, it is difficult for the classifiers to make decisions and resulting in ambiguous decision boundaries. Hence, we consider to further reinforce and align expert knowledge to the target domain through adversarial training.

**Consistency-max loss.** In traditional MCD-based methods [6, 13, 32], $L_1$ norm divergence is commonly used to measure prediction consistency between two classifiers. However, solely enforcing consistency to align different distributions is insufficient. In addition to enforcing consistency, decision boundaries should be more accurate. But it is hindered due to the lack of ground truth labels in the black-box domain adaptation setting. Fortunately, there exists a basic fact, i.e., for a given sample, the probability that both experts predict it to belong to a same incorrect category is very low. Therefore, a simple but effective divergence measurement loss is proposed. To be specific, let $P^1 = \{(p_i^1)\}_{i=1}^N$ and $P^2 = \{(p_i^2)\}_{i=1}^N$ denote the softmax predictions of the black-box guided classifier and ViL guided classifier, respectively. The Consistency-Max loss (CM loss) is defined as:

$$\mathcal{L}_{cm} = -P^1 \cdot P^2. \tag{7}$$

Next, we demonstrate the effectiveness of the proposed CM loss. For $i$-th sample, as per the Cauchy-Schwarz inequality:

$$\left(\sum_{k=1}^K p_k^1 p_k^2\right)^2 \le \sum_{k=1}^K (p_k^1)^2 \sum_{k=1}^K (p_k^2)^2, \tag{8}$$

where $K$ represents the number of classes. Since the predicted outputs are softmax probabilities, each item of the prediction has

$p_k > 0$. Thus, we can derive:

$$\sum_{k=1}^{K}(p_k^1)^2 \sum_{k=1}^{K}(p_k^2)^2 \le (\sum_{k=1}^{K} p_k^1)^2 (\sum_{k=1}^{K} p_k^2)^2 = 1. \tag{9}$$

Hence, it can be inferred as follows:

$$\sum_{k=1}^{K} p_k^1 p_k^2 \le 1. \tag{10}$$

From Eq.(10), it is evident that the maximum value 1 is achieved if and only if both classifiers assign a value of 1 to a same class for the $i$th sample. Due to the low probability of two experts predicting the $i$th sample as the same incorrect class, the predictions are more consistent and certain using CM loss constraint, which tends to be more correct.

**Adversarial alignment of experts knowledge.** We employ adversarial thought to align knowledge from different experts by leveraging the proposed CM loss. First, the discrepancy is maximized between the outputs of the two classifiers with the fixed feature extractor. Additionally, we employ experts supervision to ensure the correct classification of the classifiers:

$$\min_{h^1,h^2} \mathcal{L}_{ah} = -\mathcal{L}_{cm} + \mathcal{L}_{ce}^1 + \mathcal{L}_{ce}^2. \tag{11}$$

Next, the classifiers are fixed and only the feature extractor is updated. The discrepancy between classifiers' output is minimized as follows:

$$\min_{f} \mathcal{L}_{af} = \mathcal{L}_{cm}. \tag{12}$$

In summary, maximizing the CM loss under experts' constraints enables the identification of hard samples. Subsequently, minimizing the CM loss yields the feature extractor to learn the features that are indistinguishable by two classifiers. Namely, the extracted features are more discriminate and domain invariant. Gradually, through adversarial learning and CM loss, difficult samples are selected and further classified by the target model. From a macro perspective, the two expert knowledge are distilled to the target domain.

During the training process, we repeat the aforementioned steps until the model converges. For inference, we utilize the ViL guided classifier. The overall algorithm is outlined in Algorithm 1.

## 4 EXPERIMENTS

### 4.1 Experimental setup

**Dataset description.** We use four open datasets including Office-31, Office-Home and VisDA-2017 for comparative experiments. **Office-31** [31] is a commonly used dataset that contains three domains including Amazon (A), DSLR (D), and Webcam (W) with 31 categories. **Office-Home** [36] dataset consists of around 15,500 images spread across 65 categories, covering four diverse domains: Art (A), Clip Art (C), Product (P), and RealWorld (R). **VisDA-2017** [29] dataset comprises synthetic-to-real images. The source domain consists of 152397 synthetic images and the target domain includes 55388 realistic images.

**Model architecture.** For a fair comparison, our network employs the same structure as the state-of-the-art black-box domain adaptation methods [21, 28]. For both the source and target domains,

---

**Algorithm 1** Adversarial Experts Model for Black-box Domain Adaptation.

---

**Input**: Target domain data $X = \{(x_i)\}_{i=1}^{N}$, the target noisy labels from black-box $Y^1 = \{(y_i^1)\}_{i=1}^{N}$, pre-trained ViL model, initial prompt, initial model $\{f, h_1, h_2\}$, epoch number $T$, mini-batch number $B$.
**Output**: Adapted target model.
**Procedure**:
1: **for** $t$ = 1:$T$ **do**
2:  **for** $b$ = 1:$B$ **do**
3:    Forward a mini-batch through the target model and ViL model, getting ViL pseudo-labels $Y^2 = \{(y_i^2)\}_{i=1}^{N}$ ;
4:    **Step 1:** Update black-box guided $f$, $h_1$ by Eq.(3) and ViL guided $f$, $h_2$ by Eq.(4);
5:    **Step 2:** Update $h_1$, $h_2$ by maximizing the discrepancy and minimizing the cross entropy loss Eq.(11) ;
6:    **Step 3:** Update $f$ by minimizing the discrepancy Eq.(12) ;
7:  **end for**
8:  Update black-box labels $Y^1$ by Eq.(5) and prompt of ViL model by Eq.(6);
9: **end for**
10: **return** Adapted target model.

---

the pre-trained model ResNet-50 [11] is employed as a feature extractor for the Office-31 and Office-Home datasets, and ResNet-101 is utilized for the VisDa-2017 dataset. The same as DINE and RAIN [21, 28], between the feature extractor and the fully connected layer classifiers, a bottleneck structure is used which is a series of layers in a pipeline, consisting of a fully connected layer, batch normalization, fully connected layer, weight normalization. This structure is combined with the feature extractor as $f$ in the adversarial alignment learning. For the frozen ViL model, we utilize the widely used CLIP ViT-B/32 to obtain the output results.

**Implementation details.** Following previous works, all the black-box models are trained with the most commonly used cross-entropy loss as Eq.(1). In terms of AEM parameter, as indicated in the method part, we have only one hyperparameter, which is the momentum setting as $\alpha = 0.9$ for updating the black-box pseudo-labels. For the ViL model's prompt, we initialize it as 'a photo of a [CLS].'. The black-box pseudo-labels and the learnable prompts are updated at the beginning of each epoch. For model parameter settings, we adhere to the recommended training configurations outlined in [21, 26], encompassing weight decay (1e-3), bottleneck size (256), and batch size (64). Our optimizer employs mini-batch SGD. The same learning rate scheduler $\eta_p = \eta_0(1+\gamma p)^{-b}$ is adopted for all datasets. For Office-31 and Office-home datasets, we set $\gamma = 10$ and $b = 0.75$, while for VisDa-2017, we set $\gamma = 10$ and $b = 2.25$. All experiments are implemented on the PyTorch platform with a single NVIDIA RTX GPU.

**Competitors.** With the source (black-box) model as a baseline, we compare AEM with the state-of-the-art methods, which can be categorized into three types: (1) The first type is unsupervised domain adaptation methods, where both the source domain data and ground truth labels are accessible, such as MCD [32], AFS [46], NCL [22] , DAPL [10], AD-CLIP [33], PDA [1] . (2) The second type

**Table 1: Comparison results on *Office-31* dataset. Metric: classification accuracy (%); Backbone: ResNet-50. Comparative methods are categorized from top to bottom as baseline, UDA, SFDA, and black-box DA. '✓' in M means the usage of ViL model.**

| Method | Venue | M | A→D | A→W | D→A | D→W | W→A | W→D | *Avg.* |
|---|---|---|---|---|---|---|---|---|---|
| Source | – | ✗ | 79.9 | 76.6 | 56.4 | 92.8 | 60.9 | 98.5 | 77.5 |
| MCD [32] | CVPR18 | ✗ | 92.2 | 88.6 | 69.5 | 98.5 | 69.7 | 100.0 | 86.5 |
| NCL [22] | ACMMM23 | ✗ | 96.3 | 96.6 | 77.6 | 98.7 | 77.4 | 100.0 | 91.1 |
| DAPL [10] | TNNLS23 | ✓ | 81.7 | 80.3 | 81.2 | 81.8 | 81.0 | 81.3 | 81.2 |
| PDA [1] | AAAI24 | ✓ | 91.2 | 92.1 | 83.5 | 98.1 | 82.5 | 99.8 | 91.2 |
| SHOT [20] | ICML20 | ✗ | 94.0 | 90.1 | 74.7 | 98.4 | 74.3 | 99.9 | 88.6 |
| D-MCD [4] | AAAI22 | ✗ | 94.1 | 93.5 | 76.4 | 98.8 | 76.4 | 100.0 | 89.9 |
| C-SFDA [15] | CVPR23 | ✗ | 96.2 | 93.9 | 77.3 | 98.8 | 77.9 | 99.7 | 90.5 |
| TPDS [34] | IJCV24 | ✗ | 97.1 | 94.5 | 75.7 | 98.7 | 75.7 | 99.8 | 90.2 |
| DINE [21] | CVPR22 | ✗ | 91.7 | 87.5 | 72.9 | 96.3 | 73.7 | 98.5 | 86.7 |
| RAIN [28] | IJCAI23 | ✗ | 93.8 | 88.8 | 75.5 | 96.8 | 76.7 | 99.5 | 88.5 |
| BETA [38] | ICLR23 | ✗ | 93.6 | 88.3 | 76.1 | 95.5 | 76.5 | 99.0 | 88.2 |
| RFC [44] | AAAI24 | ✗ | 94.4 | 93.0 | 76.7 | 95.6 | 77.5 | 98.1 | 89.2 |
| AEM | Ours | ✓ | 95.1 | 94.0 | 81.8 | 98.2 | 82.6 | 99.4 | 91.9 |

**Table 2: Comparison results on *Office-Home* dataset. Metric: classification accuracy (%); Backbone: ResNet-50. Comparative methods are categorized from top to bottom as baseline, UDA, SFDA, and black-box DA. '✓' in M means the usage of ViL model.**

| Method | Venue | M | Ar→Cl | Ar→Pr | Ar→Rw | Cl→Ar | Cl→Pr | Cl→Rw | Pr→Ar | Pr→Cl | Pr→Rw | Rw→Ar | Rw→Cl | Rw→Pr | *Avg.* |
|---|---|---|---|---|---|---|---|---|---|---|---|---|---|---|---|
| Source | – | ✗ | 44.1 | 66.9 | 74.2 | 54.5 | 63.3 | 66.1 | 52.8 | 41.2 | 73.2 | 66.1 | 46.7 | 77.5 | 60.6 |
| MCD [32] | CVPR18 | ✗ | 48.9 | 68.3 | 74.6 | 61.3 | 67.6 | 68.8 | 57.0 | 47.1 | 75.1 | 69.1 | 52.2 | 79.6 | 64.1 |
| AFS [46] | ACMMM23 | ✗ | 58.7 | 80.2 | 83.3 | 67.6 | 79.0 | 76.7 | 68.9 | 57.1 | 82.6 | 75.1 | 65.5 | 85.7 | 73.4 |
| NCL [22] | ACMMM23 | ✗ | 58.9 | 78.6 | 82.6 | 69.2 | 79.4 | 78.6 | 67.2 | 57.1 | 82.3 | 73.1 | 58.7 | 85.6 | 72.6 |
| DAPL [10] | TNNLS23 | ✓ | 54.1 | 84.3 | 84.4 | 74.4 | 83.7 | 85.0 | 74.5 | 54.6 | 84.8 | 75.2 | 54.7 | 83.8 | 74.5 |
| AD-CLIP [33] | ICCV23 | ✓ | 55.4 | 85.2 | 85.6 | 76.1 | 85.8 | 86.2 | 76.7 | 56.1 | 85.4 | 76.8 | 56.1 | 85.5 | 75.9 |
| PDA [1] | AAAI24 | ✓ | 55.4 | 85.1 | 85.8 | 75.2 | 85.2 | 85.2 | 74.2 | 55.2 | 85.8 | 74.7 | 55.8 | 86.3 | 75.3 |
| SHOT [20] | ICML20 | ✗ | 57.1 | 78.1 | 81.5 | 68.0 | 78.2 | 78.1 | 67.4 | 54.9 | 82.2 | 73.3 | 58.8 | 84.3 | 71.8 |
| D-MCD [4] | AAAI22 | ✗ | 59.4 | 78.9 | 80.2 | 67.2 | 79.3 | 78.6 | 65.3 | 55.6 | 82.2 | 73.3 | 62.8 | 83.9 | 72.2 |
| C-SFDA [15] | CVPR23 | ✗ | 58.6 | 80.2 | 82.9 | 69.8 | 78.6 | 79.0 | 67.8 | 55.7 | 82.3 | 73.6 | 60.1 | 84.9 | 72.8 |
| TPDS [34] | IJCV24 | ✗ | 59.3 | 80.3 | 82.1 | 70.6 | 79.4 | 80.9 | 69.8 | 56.8 | 82.1 | 74.5 | 61.2 | 85.3 | 73.5 |
| DINE [21] | CVPR22 | ✗ | 54.2 | 77.9 | 81.6 | 65.9 | 77.7 | 79.9 | 64.1 | 50.5 | 82.1 | 71.1 | 58.0 | 84.3 | 70.6 |
| RAIN [28] | IJCAI23 | ✗ | 57.0 | 79.7 | 82.8 | 67.9 | 79.5 | 81.2 | 67.7 | 53.2 | 84.6 | 73.3 | 59.6 | 85.6 | 73.0 |
| BETA [38] | ICLR23 | ✗ | 57.2 | 78.5 | 82.1 | 68.0 | 78.6 | 79.7 | 67.5 | 56.0 | 83.0 | 71.9 | 58.9 | 84.2 | 72.1 |
| RFC [44] | AAAI24 | ✗ | 57.4 | 80.0 | 82.8 | 67.0 | 80.6 | 80.2 | 68.3 | 57.8 | 82.8 | 72.8 | 59.3 | 85.9 | 72.9 |
| AEM | Ours | ✓ | 65.4 | 88.3 | 89.5 | 80.1 | 90.7 | 89.7 | 78.9 | 61.4 | 89.9 | 79.2 | 63.6 | 90.8 | 80.6 |

is source-free domain adaptation, where only the source domain model is accessible. It contains SHOT [20], D-MCD [4], C-SFDA [15], TPDS [34]. (3) The third type is black-box domain adaptation. The only available information is the noisy labels output by the source model, DINE [21], RAIN [28], BETA [38], RFC [44] to name a few. In addition, with a checkmark '✓' in column M in all tables, it indicates the utilization of the ViL model.

## 4.2 Comparison results

The comparative results of AEM on three datasets are shown in Tables 1 to 3. Compared to other black-box domain adaptation methods, our approach emerges as a frontrunner. On Office-31 dataset, our method attains a remarkable accuracy of 91.9%, surpassing the second-highest method, RFC, by 2.7%. On Office-Home dataset, our average accuracy outperforms other leading methods by 7.6%, with optimal results across all 12 tasks. Similarly, on the VisDa-2017 dataset, we achieved the highest accuracy of 89.3%. Compared with the source-free domain adaptation methods, AEM maintains its leading position even under stringent conditions, namely without the source model. On the Office-31, Office-Home, and VisDa-2017 datasets, AEM outperforms the highest method each by 1.4%, 7.1%, and 0.6%, respectively. Moreover, the results of AEM can even match the state-of-the-art UDA methods that have access to source domain data and labels. Specifically, on the three datasets, our method out-performs the highest-performing UDA methods by 0.7%, 4.7% and 1.6%. When stacked up against methods utilizing vision-language

**Table 3: Comparison results on *Visda-17* dataset. Metric: per-class classification accuracy (%); Backbone: ResNet-101. Comparative methods are categorized from top to bottom as baseline, UDA, SFDA, and black-box DA. '✓' in M means the usage of ViL model.**

| Method | Venue | M | plane | bcycl | bus | car | horse | knife | mcycl | person | plant | sktbrd | train | truck | *Avg.* |
|--------|-------|---|-------|-------|-----|-----|-------|-------|-------|--------|-------|--------|-------|-------|--------|
| Source | – | ✗ | 64.3 | 24.6 | 47.9 | 75.3 | 69.6 | 8.5 | 79.0 | 31.6 | 64.4 | 31.0 | 81.4 | 9.2 | 48.9 |
| MCD [32] | CVPR18 | ✗ | 87.0 | 60.9 | 83.7 | 64.0 | 88.9 | 79.6 | 84.7 | 76.9 | 88.6 | 40.3 | 83.0 | 25.8 | 71.9 |
| AFS [46] | ACMMM23 | ✗ | 97.2 | 89.0 | 83.8 | 87.5 | 95.7 | 94.0 | 87.9 | 82.8 | 96.6 | 94.7 | 82.4 | 53.1 | 87.1 |
| NCL [22] | ACMMM23 | ✗ | 97.1 | 88.5 | 90.0 | 65.2 | 96.7 | 92.9 | 90.1 | 81.5 | 94.6 | 89.5 | 89.0 | 58.8 | 86.2 |
| DAPL [10] | TNNLS23 | ✓ | 97.8 | 83.1 | 88.8 | 77.9 | 97.4 | 91.5 | 94.2 | 79.7 | 88.6 | 89.3 | 92.5 | 62.0 | 86.9 |
| AD-CLIP [33] | ICCV23 | ✓ | 98.1 | 83.6 | 91.2 | 76.6 | 98.1 | 93.4 | 96.0 | 81.4 | 86.4 | 91.5 | 92.1 | 64.2 | 87.7 |
| PDA [1] | AAAI24 | ✓ | 97.2 | 82.3 | 89.4 | 76.0 | 97.4 | 87.5 | 95.8 | 79.6 | 87.2 | 89.0 | 93.3 | 62.1 | 86.4 |
| SHOT [20] | ICML20 | ✗ | 94,3 | 88.5 | 80.1 | 57.3 | 93.1 | 94.9 | 80.7 | 80.3 | 91.5 | 89.1 | 86.3 | 58.2 | 82.9 |
| D-MCD [4] | AAAI22 | ✗ | 97.0 | 88.0 | 90.0 | 81.5 | 95.6 | 98.0 | 86.2 | 88.7 | 94.6 | 92.7 | 83.7 | 53.1 | 87.5 |
| C-SFDA [15] | CVPR23 | ✗ | 97.4 | 91.8 | 87.6 | 78.1 | 96.6 | 99.3 | 90.6 | 87.2 | 95.6 | 94.6 | 88.9 | 57.3 | 88.7 |
| TPDS [34] | IJCV24 | ✗ | 97.6 | 91.5 | 89.7 | 83.4 | 97.5 | 96.3 | 92.2 | 82.4 | 96.0 | 94.1 | 90.9 | 40.4 | 87.6 |
| DINE [21] | CVPR22 | ✗ | 95.3 | 85.9 | 80.1 | 53.4 | 93.0 | 37.7 | 80.7 | 79.2 | 86.3 | 89.9 | 85.7 | 60.4 | 77.3 |
| RAIN [28] | IJCAI23 | ✗ | 96.6 | 86.8 | 83.0 | 70.9 | 94.5 | 81.8 | 84.2 | 83.6 | 90.9 | 89.5 | 89.4 | 64.0 | 82.7 |
| BETA [38] | ICLR23 | ✗ | 94.9 | 90.2 | 85.4 | 61.1 | 95.5 | 93.1 | 85.0 | 83.8 | 92.9 | 91.9 | 91.1 | 55.0 | 85.1 |
| RFC [44] | AAAI24 | ✗ | 95.6 | 89.7 | 87.8 | 75.8 | 96.5 | 96.5 | 90.4 | 82.8 | 96.0 | 70.0 | 85.7 | 55.1 | 85.2 |
| AEM | Ours | ✓ | 98.6 | 88.1 | 89.7 | 74.8 | 98.0 | 93.9 | 93.0 | 89.3 | 90.1 | 97.2 | 95.2 | 63.5 | 89.3 |

**Table 4: Ablation study on knowledge feedback strategy on tasks D→ A and W→A in *Office-31* dataset.**

| | D→ A | | | | W→A | | | |
|---|---|---|---|---|---|---|---|---|
| | $p^1$ | $p^2$ | $p^{mix}$ | Avg. | $p^1$ | $p^2$ | $p^{mix}$ | Avg. |
| w/o_bv | 68.2 | 76.4 | 74.4 | 73.0 | 69.3 | 76.0 | 75.5 | 73.6 |
| w/o_b | 68.2 | 74.0 | 72.6 | 71.6 | 70.6 | 75.8 | 74.5 | 73.6 |
| w/o_v | 73.2 | 76.6 | 75.5 | 75.1 | 75.3 | 76.8 | 77.6 | 76.6 |
| AEM | 79.0 | 81.8 | 81.6 | 80.8 | 79.5 | 82.6 | 81.9 | 81.3 |

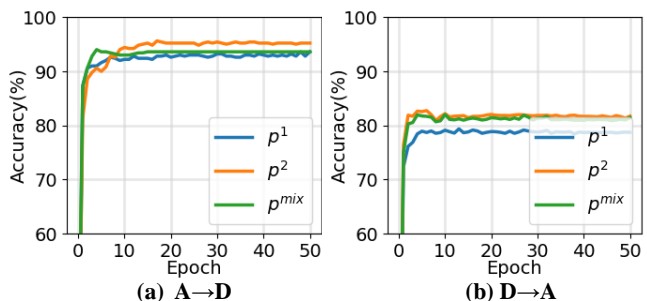

**Figure 3: Accuracy curves of (a) A→D and (b) D→A on *Office-31* dataset.**

information, such as AD-CLIP, DAPL, and PDA, which belong to UDA, our method achieves comparable results. Worth noting is that on the office dataset, AEM achieves the highest accuracy across all 12 specific tasks. We also compare our method with CLIP model and the adapted prompt CLIP model on Office-31 and Office-Home datasets, which shows AEM is better in terms of performance, model parameters and FLOPs (see Appendix).

Through the analysis of experimental data, we have the following observations. First, most of the methods only use source domain information and target data, thus failing to capture the high-level semantic information within the data. In contrast, we make use of both the source domain information from black-box expert and the high-level semantic information from ViL expert. Then, generally speaking, from the results in column 'M', it's clear that the methods, such as AD-CLIP, DAPL and PDA, leveraging ViL expert knowledge outperform those that do not. However, while they address UDA problems, we tackle a more challenging scenario of black-box domain adaptation with stringent source domain access constraints. Our method even achieves a higher average accuracy than theirs across the three datasets. This is attributed to AEM's utilization of adversarial principles. In the adversarial alignment stage, hard samples are identified through the classifiers, and the

feature extractor is improved to extract more discriminative and domain invariant features. Also, during the training, the knowledge of both experts is gradually aligned in the target domain. Furthermore, our method continuously updates the knowledge of both experts to make their guidance in the target domain more adaptable and relevant. Moreover, the proposed CM loss better measures the discrepancy between different prediction distributions, ensuring the consistency and certainty of predictions simultaneously.

## 4.3 Further studies

In order to better demonstrate the effectiveness of our method, we conduct extensive ablation analysis and visualization experiments on Office-31 dataset. The analysis on Office-Home dataset is shown in Appendix.

**Ablation study on knowledge feedback strategy.** To demonstrate the effectiveness of the proposed knowledge feedback strategy, three additional variants are compared, i.e., training without knowledge feedback strategy (w/o_bv), training without updating the black-box labels (w/o_b), and training without updating the

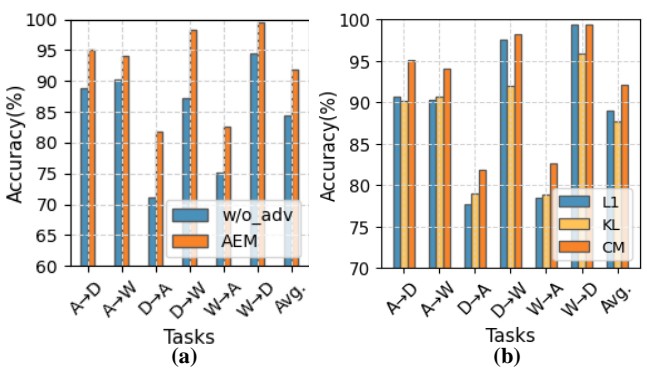

**(a)**       **(b)**

Figure 4: (a) Ablation study on adversarial learning in *Office-31* dataset. *w/o_adv* and *AEM* represent training without and with adversarial learning, respectively. (b) Accuracy using different classifier consistency loss in *Office-31* dataset.

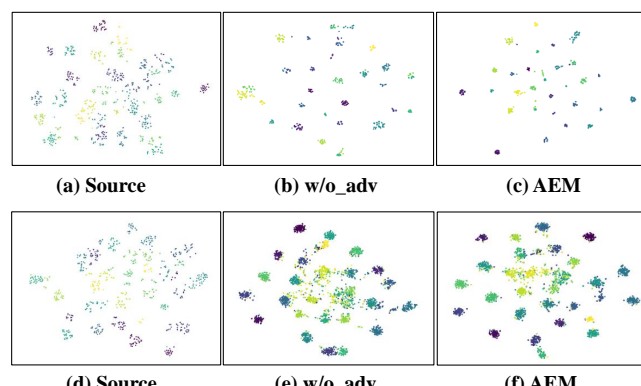

**(a) Source**  **(b) w/o_adv**  **(c) AEM**

**(d) Source**  **(e) w/o_adv**  **(f) AEM**

Figure 5: t-SNE visualizations of feature distributions learned by source model (left), w/o_adv model (middle) and AEM (right) on tasks A→D (*Top*) and D→A (*Bottom*) in *Office-31* dataset. *Zoom in for best view.*

prompt of the ViL model (w/o_v). As denoted before, $p^1$ and $p^2$ represent predictions of classifiers $h^1$ and $h^2$ guided by black-box and ViL model, respectively. $p^{mix}$ denotes the prediction obtained by averaging the outputs of both classifiers. The results are shown in Table 4, updating the knowledge of both experts yields the best results, surpassing the second highest on average by 5.7% and 4.7% for each task.

**Ablation study on different classifier combination.** Fig.3 illustrates the accuracy curve of three different predictions, i.e., $p^1$, $p^2$ and $p^{mix}$. The accuracy of these three predictions tends to be stable after 10 epochs and the differences are subtle, all around 2.0%. Overall, it appears that the classifier guided by ViL tends to perform slightly better. This is because the classifier guided by ViL contains more high-level semantic information. Therefore, in Section 4.2 and the following ablation studies, we choose the classifier guided by the ViL expert for comparison. Furthermore, as shown in Fig.3, the accuracy on each task reaches the maximum around the 10th epoch and then is stable, which means the training of the target domain has converged.

**Ablation study on adversarial learning.** To validate the effectiveness of adversarial learning, we use a consistency-based method (w/o_adv) to compare. Instead of maximizing classifier discrepancy, the method w/o_adv solely utilizes CM loss for constraint, updating both the feature extractor and the classifiers accordingly. As shown in Fig.4(a), compared to the method w/o_adv, AEM achieves an average accuracy improvement of 7.6%. This is because using consistency loss alone not only fails to address the issue of difficult samples, but also introduces noise into the target domain model due to noisy labels from black-box models. Although these noises are partially corrected by the ViL model, they also lead incorrect optimization directions to the ViL model. In AEM, we select difficult samples by maximizing classifiers' discrepancy. Then we update the feature extractor by minimizing prediction discrepancy, enabling the extracted features to be more discriminative.

**Ablation study on classifier consistency loss.** Fig.4(b) displays the results obtained using different classifier discrepancy metric loss during the adversarial alignment stage. We compare the proposed CM loss with $L_1$ loss and KL divergence loss. The

results show that using CM loss outperforms $L_1$ loss by 2.9% and KL divergence loss by 5.8% on average. The reason why KL performs worst is that it is a biased estimate. However, in the framework of AEM, the two classifiers should have equal competitive relationships. Thus the bias towards either side will result in a deterioration of model performance. $L_1$ loss performs slightly worse because it merely constrains the two predictions of the same sample to be more consistent. The proposed CM loss not only requires consistency between predictions, but also constrains the predictions to be more certain, thus demonstrating better performance.

**Visualization on t-SNE.** The t-SNE [35] visualization results on tasks A→D and D→A are shown in Fig.5. Fig.5.(a)(d) depict the visualization data directly using black-box noisy labels, presenting a scattered distribution with unclear classification boundaries. Fig.5(b)(e) displays the results without adversarial training, as described before. It can be observed that, under the guidance of two experts, the target domain model achieves better classification capability. Fig.5(c)(f) presents the results of AEM. Compared to w/o_adv, the clusters by AEM are more compact, and the classification boundaries are clearer. This is because adversarial learning helps identify hard samples and distills the knowledge from the two experts.

## 5 CONCLUSION

We propose a new framework for black-box domain adaptation based on adversarial learning, named AEM. In contrast to previous methods that exploit self-information, we introduce an additional vision-language model into black-box domain adaptation to capture high-level semantic information. Moreover, AEM is characterized by jointly leveraging the existing knowledge of black-box and ViL experts and continuously updating the experts themselves. And then, adversarial learning is employed to distill two heterogeneous knowledge to the target domain model. Moreover, the proposed simple yet effective consistency-max (CM) loss ensures the consistency and certainty of predictions. The effectiveness of our approach has been validated through experiments on multiple datasets.

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
