# OpenReview forum: "Adversarial Experts Model for Black-box Domain Adaptation"
_acmmm.org/ACMMM/2024/Conference — MM2024 Poster_

### Official Review · Reviewer_4aH9 · 2024-05-20

**Rating:** 4
**Confidence:** 3

**Summary:**

This paper proposes a domain adaptation method to achieve domain transfer through adversarial training between expert models and pre-trained models. Through the designed two-stage training method, the purpose of improving the image classification prediction performance of the final model is achieved.

**Strengths:**

The paper is well written, the description of the method is relatively clear, and a large number of experiments have been conducted. The comparison with existing methods demonstrates the effectiveness of the method proposed in this article.

**Limitations:**

However, the paper has the following shortcomings:
1. Expert models and adversarial training will inevitably increase training time. The cost of each method in model training should be reflected in the experimental section.
2. Figure 2 is confusing and needs to be redrawn to show the relationship between the methods proposed in the paper.
3. This paper uses 3 data sets instead of 4.

**Suitability:**

2

---

### Official Review · Reviewer_dE3D · 2024-05-23

**Rating:** 4
**Confidence:** 3

**Summary:**

This paper proposes a novel Adversarial Experts Model (AEM) for black-box domain adaptation. The method leverages a black-box source domain model and a pre-trained Vision-Language (ViL) model as two experts to transfer knowledge and conduct adversarial training in the target domain, improving the performance of the target domain model. The main contributions include: (1) Introducing a new adversarial learning framework that jointly exploits and refines the heterogeneous knowledge from the source domain and ViL experts. (2) Proposing a knowledge feedback strategy to dynamically update the knowledge of both experts based on target domain characteristics. (3) Proposing a simple yet effective consistency-max loss to constrain the consistency and certainty of classifier predictions. Experiments on multiple datasets demonstrate the effectiveness of the proposed method.

**Strengths:**

1. The knowledge feedback strategy to adaptively update the pseudo-labels from the black-box model and the prompt of the ViL model is a neat way to tailor the experts' knowledge to the target domain over the course of training.
2. The adversarial learning framework, consisting of maximizing classifier discrepancy to identify hard samples and minimizing it to align expert knowledge, is a principled approach.
3. The consistency-max loss is simple yet effective in enforcing both prediction consistency and certainty.
4. Extensive experiments on three datasets demonstrate the superior performance of AEM over state-of-the-art methods in both black-box DA and the more relaxed source-free DA settings. Ablation studies validate the usefulness of different components.

**Limitations:**

Weaknesses & Questions
1. The paper does not provide a clear and quantitative definition of "hard samples" that are identified in the adversarial alignment stage. Without a rigorous formulation, it is difficult to assess whether the proposed method truly focuses on the hard samples and how effectively it handles them.
2. The limitations and failure cases of AEM are not thoroughly discussed. Every method has its inherent assumptions and limitations, so it is important to analyze and present the scenarios or cases where AEM may fail or underperform.
3. In the knowledge feedback stage, could you share what the new prompt of the ViL expert looks like after fine-tuning? How does it differ from and improve upon the original prompt?
4. In the experimental results of Tables 1-3, could you analyze the reasons why AEM's advantages are more significant on Office-Home than  on Office-31 and VisDA-2017? Is it related to the characteristics of the datasets?

**Suitability:**

2

---

### Official Review · Reviewer_bdAz · 2024-05-24

**Rating:** 3
**Confidence:** 2

**Summary:**

This paper proposes an Adversarial Experts Model (AEM) for black-box domain adaptation, which leverages vision-language multimodal models to incorporate rich semantic information. The target domain model consists of a feature extractor and two classifiers, trained over two stages: knowledge transferring and adversarial alignment. The black-box source model and the ViL model act as experts, guiding the learning of the classifiers while being updated to address their limitations. Adversarial learning and a novel consistency-max loss function are employed to distill expert knowledge and improve prediction certainty. Experiments on multiple datasets demonstrate the effectiveness of the proposed approach.

**Strengths:**

a. Novel adversarial learning framework leveraging heterogeneous knowledge from two experts (black-box source model and ViL model) for accurate information transfer.
b. Refinement of expert predictions by updating black-box labels and ViL model prompts, allowing knowledge to evolve and better suit the target domain.
c. Improved prediction consistency and certainty through a novel consistency-max (CM) loss, addressing limitations of previous approaches.
d. Comprehensive experiments on multiple datasets demonstrating the effectiveness of the proposed approach in various black-box domain adaptation scenarios.

**Limitations:**

Upon reviewing the comparative experiments, it has been noted that the PDA method utilized both ResNet50 and ViT-B/16 as backbone architectures. However, when presenting the performance comparison, it appears that only the results of PDA with the ResNet50 backbone were included. To provide a more comprehensive evaluation and convincingly demonstrate the effectiveness of the proposed adversarial approach, it is strongly recommended to conduct experiments using the same ViT-B/16 backbone for PDA and report the complete results. This will ensure a fair comparison and allow readers to assess the true impact of the adversarial learning framework on the black-box domain adaptation task.

**Suitability:**

2

---

### Meta-Review · Area_Chair_RZvv · 2024-07-07

**Recommendation:** Accept (Poster)
**Confidence:** 3

**Metareview:**

This paper deals with a relatively less studied and interesting problem: black-box domain adaptation. In this problem, the source domain model is wrapped as an API instead of a trainable model. And the source data is also inaccessible.
To tackle this challenging scenario, this paper proposes to use pre-trained Vision-language models as the ViL expert and enforce interactions and constraints between this new ViL expert and existing Black-box source expert. Specifically, two training stages are designed: knowledge transferring and adversarial alignment (better visualized in Rebuttal Figure 2 than the main text).
In those two stages, knowledge feedbacks and consistency-max loss are novel technical modules for the black-box domain adaptation problem, for correcting noisy labels and aligning different experts.
Good experiments including SOTA comparison, ablation, and visualization are demonstrated.

From intial comments, reviewer bdAz raised the concern of backbone (ResNet50 and ViT-B/16), i.e., baselines such as PDA mostly used ResNet while the proposed method utilized ViT-B/16 (unfair comparision). In the rebuttal, the authors made a comprehensive rebuttal and explanation (including previous lack of clarification issue: "PDA using ViT-B/16" in Table 1 of main text). Moreover, ViT-B/16 comparison is conducted on Table 1 of Rebuttal, indicating slightly improvement of proposed AEM over PDA (89.9vs89.7). Considering the difficulty of black-box DA over UDA, this is meaningful to me. Reviewer bdAz kept the score as **Borderline Reject** after rebuttal, but after carefully reading the paper and rebuttal, I think the concern is addressed.
Other two reviewers raised concerns about technical details, Figure 2 presentations, failure cases etc. In the rebuttal, the authors tried to address these concerns. These two reviewers are happy with the rebuttal, and lean towards **Borderline Accept** the paper.

Finally, the decision points are: (1) the problem and proposed solution is less investigated and meaningful. The audience will benefit from this paper. (2) The overall writing and organization (after revised Figure 2 in the rebuttal) are well and clear, which makes the paper easy to read and follow. (3) The authors clarified technical details and addressed most of the concerns.

Overall, I would like to recommend **Accept (Poster)** for this paper. Meanwhile, I also encourage the authors to include the clarification and revision of the rebuttal in the revised paper.